# Automated Monitoring of Bluefin Tuna Growth in Cages Using a Cohort-Based Approach

Pau Muñoz-Benavent [1,*], Gabriela Andreu-García [1], Joaquín Martínez-Peiró [1], Vicente Puig-Pons [2], Andrés Morillo-Faro [2], Patricia Ordóñez-Cebrián [3], Vicente Atienza-Vanacloig [1], Isabel Pérez-Arjona [2], Víctor Espinosa [2] and Francisco Alemany [4]

[1] Institute of Control Systems and Industrial Computing (AI2), Universitat Politècnica de València (UPV), 46022 Valencia, Spain; gandreu@upv.es (G.A.-G.); joama14j@upv.es (J.M.-P.); vatienza@upv.es (V.A.-V.)
[2] Institut d'Investigació per a la Gestió Integrada de Zones Costaneres (IGIC), Universitat Politècnica de València (UPV), 46730 Gandia, Spain; vipuipon@upv.es (V.P.-P.); anmofa@upv.es (A.M.-F.); iparjona@upv.es (I.P.-A.); vespinos@upv.es (V.E.)
[3] Zunibal S.L., Idorsolo Kalea, 1, 48160 Derio, Spain; patricia.ordonez@zunibal.com
[4] Atlantic-Wide Research Programme for Bluefin Tuna (GBYP), International Commission for the Conservation of Atlantic Tunas (ICCAT), 28002 Madrid, Spain; francisco.alemany@iccat.int
* Correspondence: pamuobe@upv.es

**Abstract:** In this article, the evolution of BFT (bluefin tuna) sizes in fattening cages is studied, for which it was necessary to perform exhaustive monitoring with stereoscopic cameras and an exhaustive analysis of the data using automatic procedures. Exploring the size evolution of BFT over a long period is an important step in inferring their growth patterns, which are essential for designing smart aquaculture and sustainable fishing, and even assessing their health status. An important objective of this work was to verify whether tuna in captivity, in addition to fattening, grow in length. To this end, our autonomous monitoring system, equipped with stereoscopic cameras, was installed from 28 July 2020 to 23 May 2021 in a fattening cage in the Mediterranean containing 724 free-swimming tuna. This system provides thousands of images that, grouped by time intervals, allow us to conduct our studies. An automatic procedure, already introduced in a previous work and capable of processing large volumes of data, is used to estimate the length and width of individuals in ventral stereoscopic images of fish, and the evolution over time is analysed for each biometric characteristic. However, verifying the evolution of length and width based only on means or medians of these measurements may be inconsistent and insufficiently accurate to support our study objectives, as individuals of different sizes and ages may grow at different rates. Therefore, a modal analysis (Bhattacharya's method) was undertaken to identify the cohorts within the population. The results showed that each modal length surpassed the length of the next cohort and that there was accelerated growth in cages compared to the wild. In addition, we proved that using a length–width–weight relationship to estimate fish weight gives more accurate results than traditional length–weight relationships for fish fattened in cages.

**Keywords:** bluefin tuna growth; fish monitoring; fish weight estimation; stereoscopic computer vision; cohort-based approach

**Key Contribution:** An automatic procedure is used to estimate the length and width of individuals in stereoscopic images of fish, and the evolution over time is analysed. We show that fish experience accelerated growth in cages compared to the wild, as demonstrated by applying a modal analysis (Bhattacharya's method) to identify the cohorts in the population. Furthermore, using a length–width–weight relationship to estimate fish weight gives more accurate results than traditional length–weight relationships.

## 1. Introduction

Our motivation in conducting this research was to solve specific challenges that hinder sustainable, intelligent, and precise aquaculture by combining stereoscopic vision techniques and automatic computer vision methods that allow us to extract information from large volumes of data while fish are swimming freely. The large dimensions of oceans and seas make observing and monitoring the marine environment a titanic job. However, many countries are proposing sustainability guidelines and as part of those, they are making these efforts in specific ecosystems, even though they require great human and technological input. Recently, fish farmers, ecologists, and governments have remarked on the urgent need to accurately estimate both the biomass of schools and individual fish in their natural environment [1]. With that goal, the collection of numerous accurate data on fish size or age without the need to physically handle live fish has been identified as an essential requirement. Traditional methods based on manual measurements are invasive, expensive, and stressful for animals, which limit the amount of data collected, reducing the validation of the tests performed. Instead, the quantitative estimation of fish biomass forms the basis of scientific fishery management and conservation strategies [2,3]. Biomass estimations are an important input for the design of adequate models for the assessment of fisheries, to explore fish growth stages, define growth models, and evaluate the health status of the fish. Species such as tuna and salmon are the most commonly farmed due to their market acceptance and rapid growth [4]. Bluefin tuna (BFT) is one of the most in-demand fish species in the world. Currently, the need to strengthen its management has increased research on BFT aquaculture production in many countries [5].

The ICCAT (International Commission for the Conservation of Atlantic Tunas) recommends using stereoscopic vision systems to size live fish in order to control catches for tuna farming. Commercial stereoscopic systems such as AQ1 AM100 [6] and AKVAsmart, formerly VICASS [7], require human intervention to mark the snout and fork of each fish in the image, from which to deduce the fish length and weight. This slows the process, makes it laborious, introduces the variability of manual measuring, and limits the number of samples that can be gathered to statistically represent the fish stock. The need to develop fully automatic solutions has been pointed out in [8,9], among others. In these solutions, fish weight may be deduced from fish length, as established in [10]. However, recent works [11,12] have attempted to find better approximations for their weight by using both fish length and width measurements. They work on the hypothesis that a better estimation of the biomass can be achieved by using more dimensions of the tuna than only length.

A significant contribution towards a commercial system for fully automatic fish sizing using stereoscopic vision was proposed in [13], based on geometric models of tuna that are adapted to the fish silhouette while swimming, which can provide accurate measurements of fish length in adult tuna. An extension of this proposal is presented in [14], in which new features are added to the deformable model of tuna so that it is capable of estimating the width of individuals in ventral-perspective images. In their conclusion, the authors comment that the system could be used to track the growth of fish over time by programming a recording schedule, and it could be integrated into an autonomous monitoring system, whose computing performance and energy requirements should be dimensioned to allow the recording and analysis of a statistically representative number of measurements.

Working with stereoscopic video requires the joint processing of two frames for each instant of time acquired. In addition, the fish, the object we wish to detect and measure, are swimming freely, which means we do not know when they are in front of our acquisition system. It is also unknown whether we are acquiring the best pose to estimate the fish's size. Therefore, long-term continuous monitoring of fish is necessary to obtain an adequate number of frames to take measurements. A complex sequence of steps and essential procedures, such as camera calibration, the acquisition of hours of video, video preprocessing, object detection, instance segmentation, stereoscopic correspondence in pairs of images, and 3D triangulation, must be usually performed to estimate individual measurements.

Against that background, this paper presents the findings of the BFT Growth in Farms Pilot Study (ICCAT GBYP 10/2020) and reports on the evolution of length and width distributions. The project's full report is accessible online [15]. Measurements were estimated using software that automatically processes data collected from a stereoscopic vision system from a ventral perspective (sensors placed at the bottom of the cage, facing towards the surface). This work validates the computer vision procedure developed by the authors and previously presented in works [13,14] as a valuable tool to study the growth of BFT in terms of length and width within farms. Our study offers valuable data on tuna growth in captivity, of significant interest to both fish farmers and researchers, filling a gap that has persisted until now. The large number of monitored day, as well as the extensive period of months required considerable effort but simultaneously enhanced the validity of the results. Specifically, we investigated the hypothesis of accelerated growth in cages and found an affirmative answer to the following question: Does annual growth in cages surpass that in the wild due to the special conditions within the cage, as previously demonstrated in juvenile BFT [16]? Using a cohort-based approach, we demonstrated that each modal length surpassed the length of the next cohort, indicating accelerated growth in cages compared to the wild. In addition, we established that using a length–width–weight relationship to estimate fish weight yields more accurate results than traditional length–weight relationships.

## 2. Materials and Methods

### 2.1. Description of the Subsea Monitoring System

The monitoring system comprises a subsea sensor platform equipped with a stereoscopic camera, an inclinometer, and a 120 kHz single-beam transducer. This platform was positioned at a depth of 23 m on the bottom of a fattening cage in Grup Balfegó, located in the West Mediterranean (40°51'33'' N 0°50'59'' E). Additionally, a logging subsystem, tethered to the cage structure, was placed at the water's surface. The cage itself is cylindrical, with a diameter of 50 m at the water surface and a height of 35 m at its lowest point, providing an approximate volume of 20,000 m$^3$ and accommodating 724 BFT. The autonomous monitoring system was installed and operational from 28 July 2020 to 23 May 2021. Recorded data were uploaded to the cloud on a daily basis and processed offline. Figure 1 provides an overview of the system.

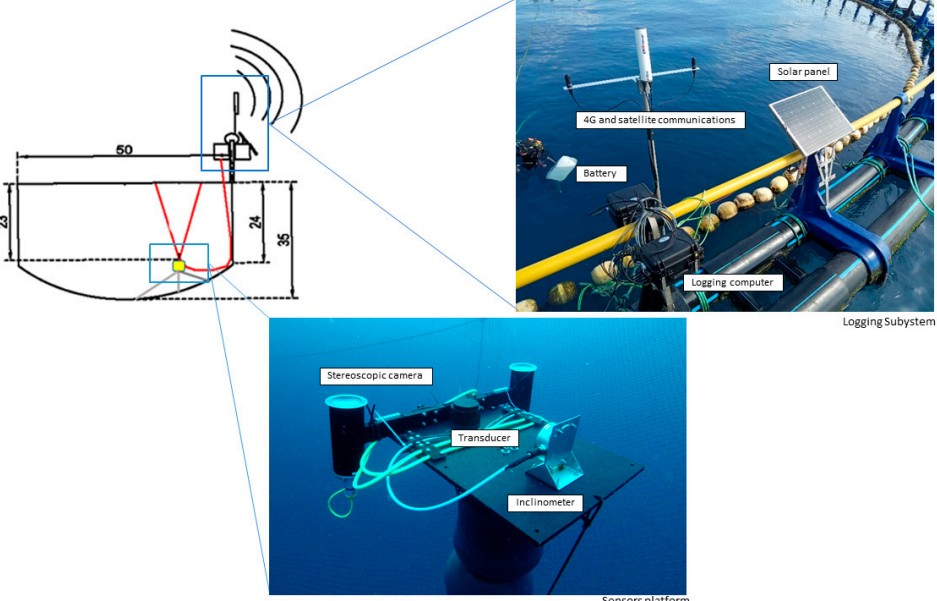

**Figure 1.** Overview of the components of the monitoring system installed at the Grup Balfegó facilities, consisting of a sensor platform and logging subsystem.

The sensor platform was equipped with a stereoscopic camera, among other sensors. Positioned on the bottom of the cage and oriented towards the surface, the camera provided a ventral perspective of the fish. Buoyancy was achieved using buoys, and ropes tied the platform to the cage. This camera arrangement offered two primary advantages: first, the sunlight acts like a backlight system so objects are always darker than water and no additional lighting equipment is required at a depth of 23 m; second, the orientation allowed for clear observation and analysis of fish body bending.

Video recordings were captured using a customised stereo camera comprising two Gigabit Ethernet cameras, each with a resolution of 2048 × 1536 pixels (3.1 megapixels) and a framerate of 20 fps. The cameras were mounted in an underwater housing, with a baseline of 85 cm and inward convergence of 5°. Camera synchronisation was achieved using the IEEE 1588 Precision Time Protocol (PTP). The system was designed for a depth of 40 m and had an umbilical cable supplying power over Ethernet to the cameras and transferring images to a logging computer, which encoded left and right videos through hardware encoding. Two systems with different focal lengths were employed. Initially, a system equipped with 12.5 mm focal length lenses was deployed. However, during the initial months, it became apparent that water turbidity conditions were considerably worse than those observed in l'Ametlla de Mar. Consequently, on many days, tuna could not be effectively measured across the entire water column. With the 12.5 mm focal length lenses, fish could only be accurately measured at distances starting from 6 m since they could not fit completely into the field of view at shorter distances. As a response to this challenge, in September 2020, we decided to modify the cameras and replace the lenses with 6 mm focal length lenses. This adjustment allowed for effective sizing of the fish at distances as close as 3 m, addressing the limitations encountered with the initial lens configuration.

The logging subsystem comprised the following elements: battery, solar panel, logging computer, satellite communication (Iridium), and 4G communications. The battery and solar panel collectively provided the system with an average daily energy autonomy of 5 h from June to October and 3 h from October to January. Iridium communications facilitated remote on/off switching and 4G communications enabled the cloud storage of the recordings.

Before installation at the Grup Balfegó facilities, the stereoscopic system underwent calibration in a controlled environment—a swimming pool measuring 12 × 6 × 2 m. A 1.40 × 1.10 m checkerboard pattern was systematically moved from −45° to 45° with respect to the optical axis and placed at distances ranging from 1 to 10 m away from the cameras. The MATLAB Stereo Calibration Application, based on [17,18], was used to accurately estimate the calibration parameters. Images captured by the stereoscopic system are presented in Figure 2.

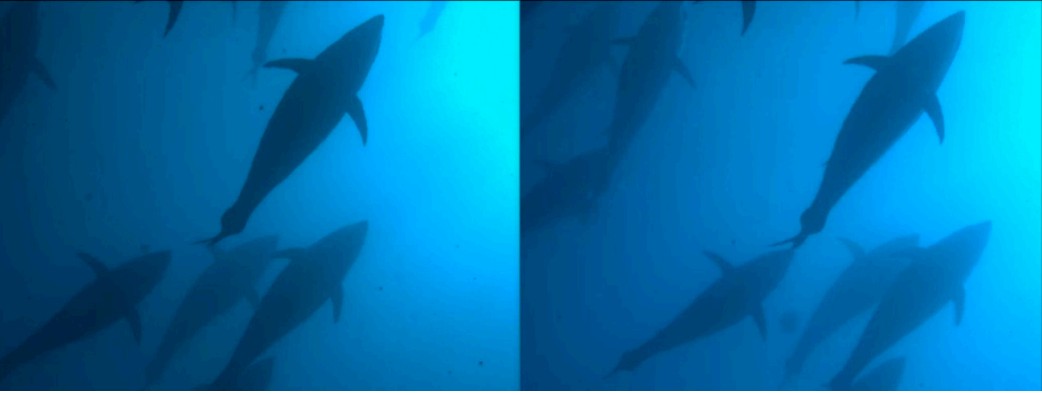

**Figure 2.** Stereoscopic images acquired with the monitoring system.

### 2.2. Computer Vision Algorithms for Fish Width and Length Estimation

The estimation of tuna sizes using computer vision techniques is based on fitting a deformable model of the fish's ventral silhouette. The algorithms involved in the process of fish sizing are summarised in Figure 3. Image segmentation is implemented using local thresholding [19], a region-based technique for extracting compact regions (blobs) on each video frame, coupled with morphological operations. The segmented blobs are geometrically characterised and sifted using shape (aspect ratio), pixel density, and dimensional filters. An edge detection algorithm is then applied, and a minimisation algorithm is used to fit a deformable tuna model. To evaluate the goodness of the fit, a Fitting Error Index (FEI) is computed. The FEI, based on the quadratic distance between the model points and target edges points, ranges from 0 to 10, where FEI = 0 denotes a perfect fit between the segmented blob and the geometric model. Fittings with high values (FEI > 3) are discarded. The results from left and right videos, obtained separately, are merged to calculate straight fork length (SFL) and the maximum width of the fish. The image plane information is then transformed into 3D measurements using the calibration parameters of the stereoscopic vision system and 3D triangulation. Samples are excluded if stereo correspondence is not met for the first and last model vertebrae, that is, if the distance from the points to the epipolar lines exceeds a defined threshold. As fish are deformable due to swimming motion, measurements from a single frame may lack reliability [8]. Two common approaches are used in the literature to mitigate the effect of swimming motion on length measurement: (i) taking measurements in all frames and deducing straight body length from a sinusoid-like pattern [8]; (ii) accounting for body bending by adding contiguous linear segments [20]. In this study, the swimming length problem was addressed by considering the tuna model bending angle θ. Valid samples were identified as those where vertebral points formed a straight line, and others were discarded. The challenge of not observing the tail fork from the ventral perspective was tackled by excluding the caudal fin from the deformable tuna model. The relationship SFL = 1.0312 ML + 0.065641, deduced from experimental samples in [13], where ML is the model length, was then applied. The system automatically provides length and maximum width measurements of tuna, contributing to a more accurate estimate of their weights. See Refs. [13,21] for further details of the computer vision algorithms.

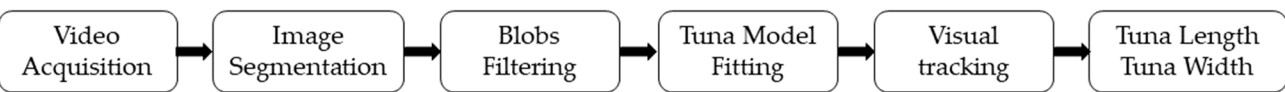

**Figure 3.** Sequence of processes performed automatically in our proposal.

### 2.3. Bhattacharya's Method for Modal Analysis

Taking into account that the fish stock comprised individuals from various age groups, a modal analysis was essential to identify different cohorts before analysing the evolution of length and width. The modal analysis outlined in this section follows the recommendations of [22], specifically applying Bhattacharya's method [23] through FiSAT II (FAO-ICLARM Fish Stock Assessment Tools) Version 1.2.2 software [24]. This method involves the separation of normal distributions, each representing a cohort of fish, from the overall distribution.

In spite of the usefulness of objective criteria in allowing researchers to conduct effective and standardised analyses, FiSAT II software allows the operator to decide on the selection of points defining each modal group, and Bhattacharya's method is sensitive to such selections. Therefore, different results may be obtained depending on operator decisions. In this case, as a first step to obtaining reliable results, available information on BFT growth rates, both in the wild and under farming conditions [16,25–36], were analysed to obtain reference values regarding the expected amplitudes of annual cohorts and the separation index between them throughout the life cycle. Moreover, to ensure the coherence among different analyses, in addition to the basic criteria to identify and characterise modal groups stated in [24], the following criteria were strictly applied:

- SI (separation index) values between successive cohorts should be similar in different analyses, with a minimum value of 2 considered as a general reference, occasionally accepting slightly lower values in older cohorts.
- SD (standard deviation) values of identified modal groups should fall within the range of 3 to 7 cm, considering background information about the variability within annual cohorts.
- The proportions of specimens belonging to given cohorts should be consistent among different analyses.

Prior to applying the method, length–frequency data underwent preprocessing with different configurations regarding class intervals and smoothing. The bin width or class interval, which represents the distance between classes or subintervals of the data, was set to 2 cm, 3 cm, or 5 cm, whereas data smoothing was applied using running averages over 3 and 5 classes. Based on preliminary analyses considering the available knowledge on Atlantic BFT annual growth, the optimal configuration for characterising annual cohorts was determined to be a bin width of 3 cm and applying a running average over 3 classes to smooth the data, since larger bins tend to mix several annual cohorts within the same modal group. Hence, the modal analysis presented in the Results section used these preprocessing options.

## 3. Results

As mentioned in Section 2.1, the autonomous monitoring system was operational from 28 July 2020 to 23 May 2021 within the fattening cage, capturing a ventral perspective of the fish. The system is able to provide thousands of accurate automatic measurements per day under ordinary circumstances. However, different factors impede its consistent functioning, primarily including limited visibility and component damage resulting from adverse weather conditions or prolonged exposure to the marine environment. Given these challenges, the recorded data were organised into consecutive-day periods, as detailed in Table 1, with a maximum interval of five days between them. This approach allowed for the accumulation of thousands of measurements within each period.

**Table 1.** Recordings with the automatic stereoscopic system grouped into consecutive-day periods.

| Year 2020 | 28–29 July | 8–11 August | 23–26 September | 10–14 October | 31 October–3 November | 15–19 November | 6–8 December | 15–18 December |
|---|---|---|---|---|---|---|---|---|
| Year 2021 | 21–22 January | 15 February | 24–26 March | 5–7 May | 21–23 May | | | |

### 3.1. Straight Fork Length (SFL) Evolution

Exploring the evolution of tuna lengths over an extended period is an important step in understanding their growth patterns and even assessing their health status. In our initial approach to analyse this length evolution, we estimated the temporal evolution of the mean length within the caged population. Figure 4 illustrates this evolution using a boxplot of SFL measurements. In each box, the lower side of the central rectangle represents the 25th percentile, whereas the upper side represents the 75th percentile. The red segment inside the rectangle indicates the median. Notably, the evolution of the median SFL appears inconsistent, showing no discernible progression over time.

Subsequently, a second approach was applied with the aim of achieving more accurate results regarding the growth rates of different cohorts within the caged population. This approach relied on the use of Bhattacharya's method described in Section 2.3, applied to the SFL measurements. The outcomes of the modal progression analysis, derived from the application of Bhattacharya's method, are presented in Table A1 in the Appendix A. It is noteworthy that the percentage of measurements and the number of individuals for certain cohorts, specifically the first and last ones (comprising smaller and larger specimens), are relatively low. This occurrence is likely due to the scarcity of specimens with those particular sizes in the monitored cage. Although these fish belong to cohorts distinct from

the well-represented ones, the limited number of individuals precludes fitting the data into a modal group. In other instances, defining a modal group with a small number of specimens is feasible, but the mean lengths of such cohorts may not be accurately represented. Therefore, despite their inclusion in the dataset, these cohorts were not considered for modal progression analyses.

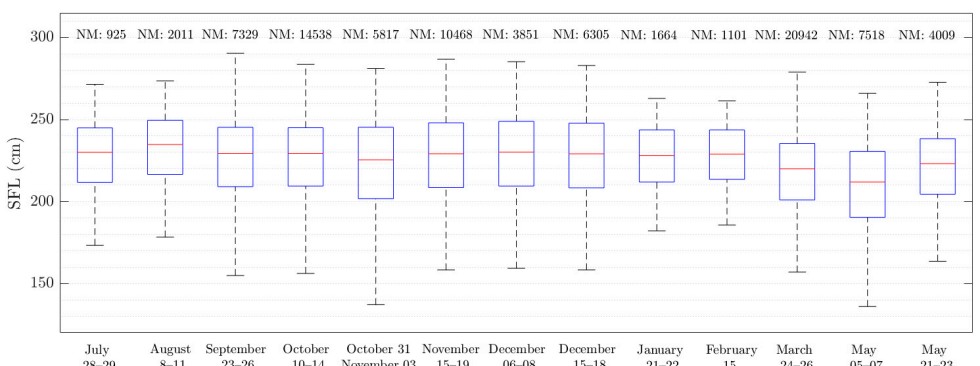

**Figure 4.** Boxplot of the evolution of straight fork length (SFL) measurements. NM: Number of measurements.

The average SFL for each period and its temporal evolution can be observed in Figure 5 and Table A2 in Appendix A (excluding cohorts with few specimens). The results indicate that the length growth from July 2020 to May 2021 ranged approximately between 11 and 26 cm, depending on the initial fish length. The decreasing separation between cohorts aligns with expectations, as annual growth rates typically decrease with age. It is also expected that after one year, the growth of each cohort will either be similar to the distance between consecutive cohorts of the same size range in the initial length distribution or even higher, because of the special conditions in a cage compared to the wild. Our results show that by May 2021, the modal lengths reached or slightly exceeded the length of the next cohorts in July 2020. This suggests that, after one year, each modal SFL surpassed the SFL of the next cohort, indicating accelerated growth in cages, as hypothesised. Variations in the growth tendency in January, February, and early May 2021 can be attributed to the impact of water turbidity on the distance at which fish could be measured. The mean distance (Z) decreases in those samples from 8 to 6 m. Additionally, note that Z is higher in the initial months due to the use of 12 mm focal length lenses, as described in Section 2.1.

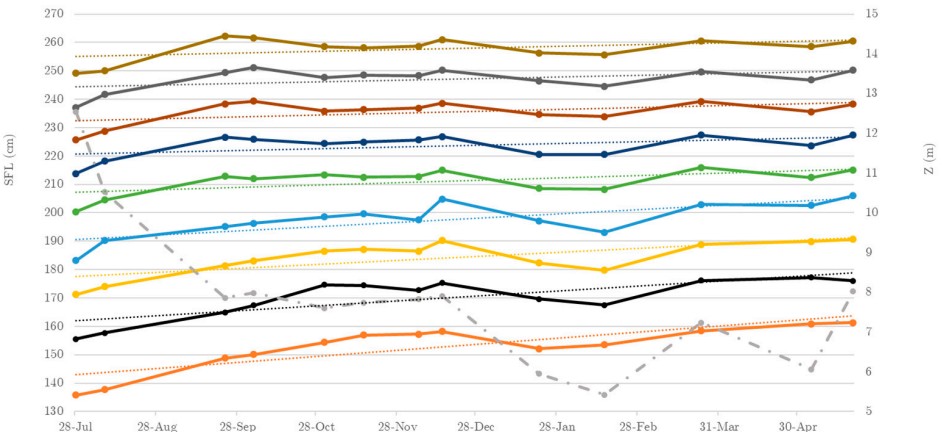

**Figure 5.** Identification of cohorts and evolution of average straight fork length (SFL) resulting from Bhattacharya's method. Tendencies are represented by dashed lines. Each line represents a cohort from cohorts 2 to 10. Cohorts 1 and 11 with few specimens have been omitted. Mean distance (Z) from the stereocamera to the fish in the measurements obtained over time is depicted as a dash-dotted line in the secondary axis.

This finding challenges the conclusions drawn in one of our previous studies [13], where an analysis based on the evolution of means led to the incorrect conclusion that fish length growth was minimal from July to October in fattening cages.

To substantiate the likelihood that the cohorts identified in July 2020 corresponded, at least partially, to annual cohorts, we compared our modal lengths with the expected body lengths as a function of age according to the well-known Von Bertalanffy growth equation applied to BFT [25,26,28–31]. As depicted in Figure 6, the modal lengths aligned with the growth equations, suggesting that they likely represented annual cohorts between 5 and 16 years old.

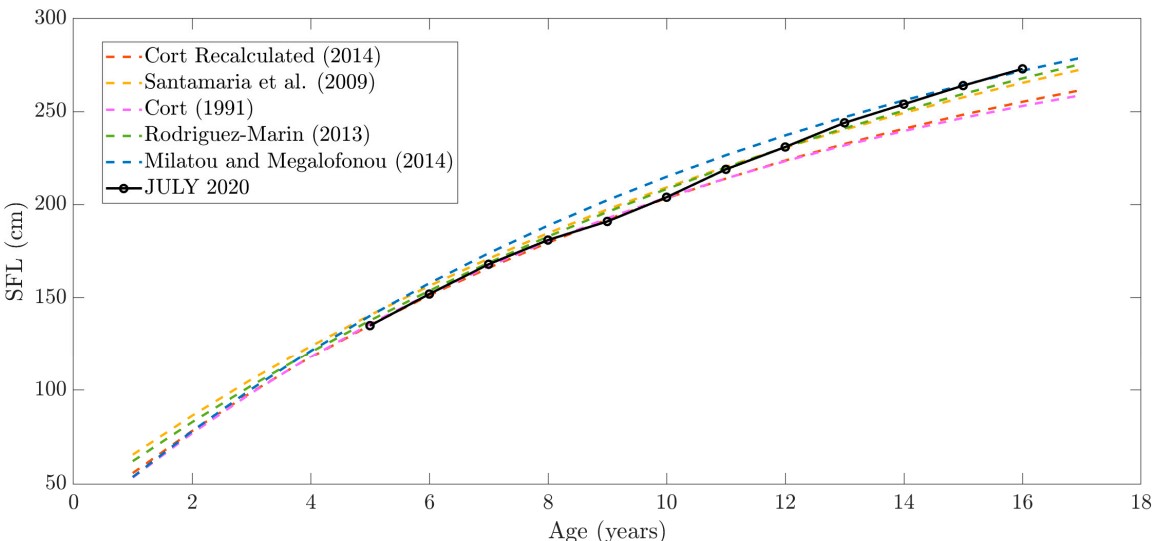

**Figure 6.** Comparison of cohorts identified in July 2020 using the automatic system and Von Bertalanffy growth curves for BFT [25,26,28–31] from various studies, aimed at corroborating the correspondence of the cohorts identified in July 2020 with annual cohorts.

### 3.2. Width Evolution

Deformable model fitting, as described in Section 2.2, allows for the estimation of fish width, defined at the thickest part of the fish silhouette, particularly when observed from a ventral perspective. When examining the frequency histogram in Figure 7a, it appears that the distribution shifts towards larger widths (A) from July 2020 to May 2021, i.e., suggesting an increase in fish width. However, given the presence of different cohorts, it is advisable to study width variations within groups of SFL. Additionally, the width increase is evident in the scatter plots of Figure 7b, where, in May 2021, the points reach higher values of A for the same SFL. It is notable that the point cloud is narrower in July 2020 and wider in May 2021, indicated greater variability in widths, implying that some fish experienced more substantial fattening than others.

Given the presence of distinct fish cohorts, the study of width involves grouping specimens with similar SFLs. Thus, SFL was segmented into groups of 10 cm, ranging from 140 to 280 cm. The average width ($\overline{A}$) for each group was then calculated at different checkpoints spanning from July 2020 to May 2021. The results, presented in Table A3 in Appendix A and depicted in Figure 8, revealed an increase ranging from 3.0 to 8.0 cm (equivalent to 9% to 17%), depending on the fish's SFL.

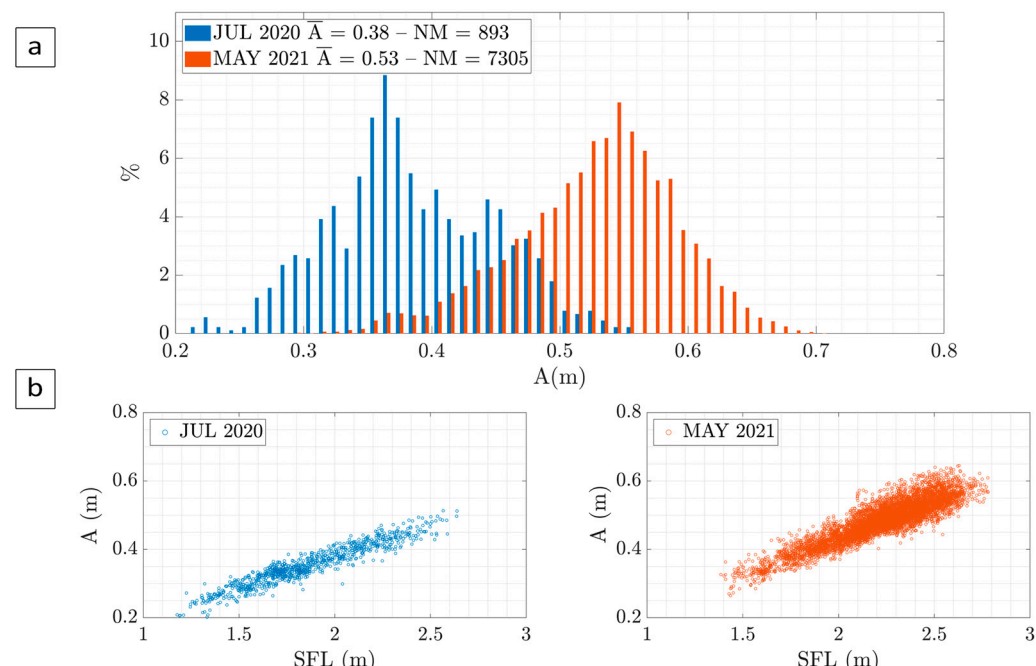

**Figure 7.** (**a**) Width (A) frequency histograms for July 2020 and May 2021. (**b**) Scatter plots of straight fork length (SFL) and width (A) for July 2020 and May 2021.

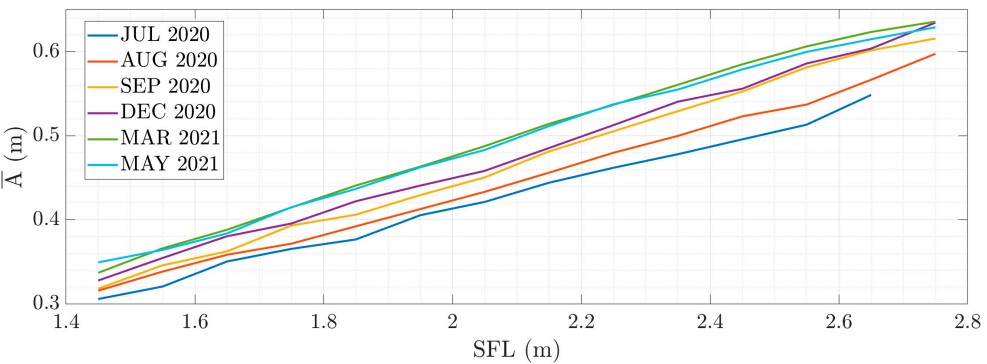

**Figure 8.** Average width ($\overline{A}$) for fish grouped according to their straight fork length (SFL) between July 2020 and May 2021.

### 3.3. Condition Factor Evolution

As the change in width appears to be proportionally consistent across different length groupings, this section examines the condition factor, defined as the ratio of the fish's width to its length. Utilising this factor removes the need for length groups, enabling a more distinct separation of the fattening versus growing aspects of the operation.

In Figure 9, a four-stage evolution of the condition factor can be observed: a significant increase during the initial three and a half months in cages (28 July 2020 to 15 November 2020), followed by a plateau in the subsequent two months (15 November 2020 to 22 January 2021), another increase in the subsequent two months (22 January 2021 to 24 March 2021), and another plateau in the last two months (24 March 2021 to 21 May 2021). When compared to environmental variables and the amount of feed, the increase in the first stage and the plateau in the second stage appear directly correlated with water temperature and feed quantity. However, the increase in the third stage and the plateau in the fourth stage do not exhibit the same direct relationship. In this case, the relationship could be elucidated by considering the dissolved oxygen concentration and the amount of feed. The increase in the factor aligns with an increase in dissolved oxygen concentration, whereas the plateau coincides with a decrease in the amount of feed. Additionally, it is known that

from January onwards, when daylight begins to increase, physiological changes associated with the annual reproductive cycle occur. It is important to note that the feeding regime adheres to the company's production policy and was not part of the experimental design.

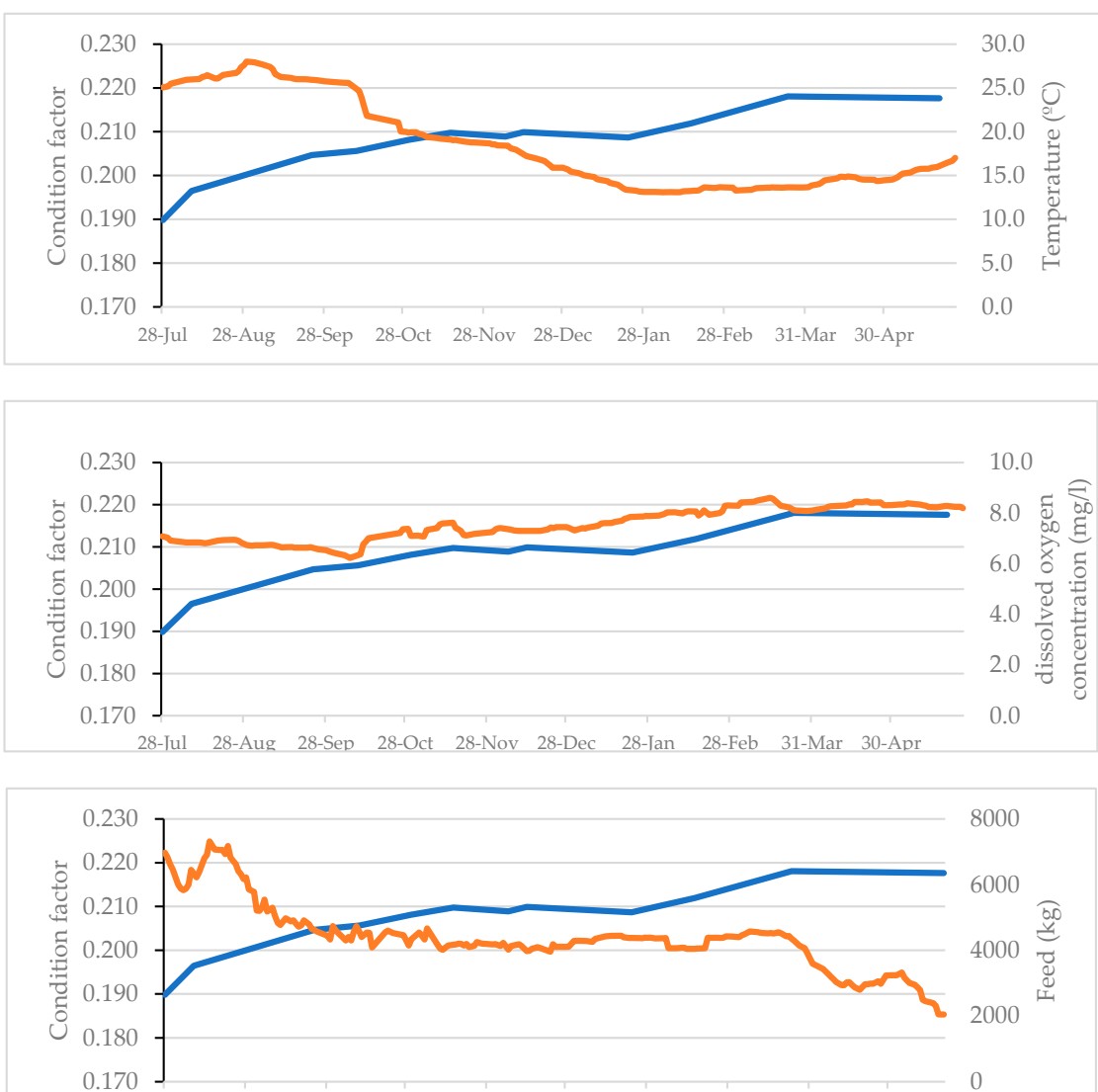

**Figure 9.** Evolution of the condition factor (in blue) with different environmental variables (water temperature and dissolved oxygen concentration) and amounts of feed (in orange).

### 3.4. Tuna Weight Estimation

The ICCAT proposes the use of the expression proposed in [10] to estimate tuna weight (W) from tuna length (SFL), which demonstrated good fitting accuracy during the purse seine fishing season in the Mediterranean:

$$W = 2.8684 \cdot 10^{-5} \cdot SFL^{2.907}, \tag{1}$$

However, in [11,12], relationships between weight and linear dimensions of BFT were analysed using data from Grup Balfegó harvests. One of the conclusions drawn was that weight could be more accurately estimated by considering various dimensions apart from length. The following equation was proposed for use when both length and width (A) are available:

$$M3 = 7.21719 \cdot 10^{-5} \cdot SFL^{2.07092} \cdot A \tag{2}$$

Considering that the automatic system can provide both length and width measurements, the weight of each sample can be estimated using Equation (2). The accuracy of each equation was previously examined in [11,12], yielding a mean relative error of 25.68% for Equation (1) and 4.88% for Equation (2).

Between March and May 2021, 645 out of the 724 BFT (89%) were harvested, and their SFL and weight were recorded at Grup Balfegó. In Figure 10, the weights from harvests and the estimations using the SFL-W relationship from [10] are presented, along with the weights estimated using the M3 equation and the automatic system measurements in the last recordings before harvesting (21–23 May). As depicted in Figure 10, the weights estimated from SFL measured at harvesting differ from the weights at harvesting after the fish has remained in the fattening cages for some months. Nevertheless, the weights estimated with the M3 equation and the automatic system measurements align well with the measured weights at harvesting. Moreover, as illustrated in Figure 11, the weights in May 2021 are higher than in July 2020 for the same lengths, confirming the greater fish growth in weight under farming conditions compared to that in the wild.

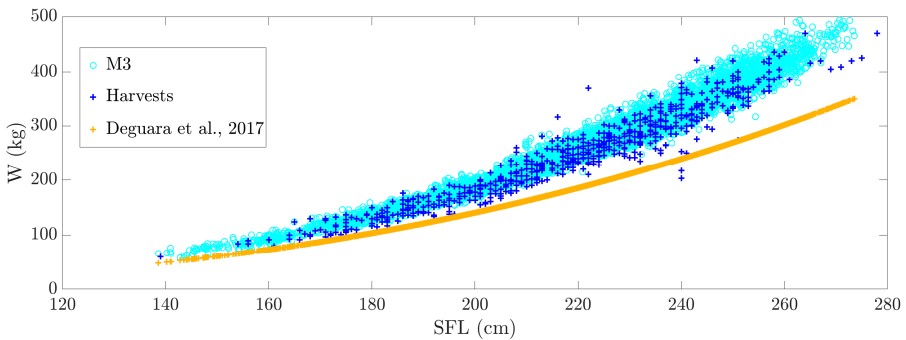

**Figure 10.** Comparison of measured weights (W) at harvesting (May 2021), estimated weights using SFL-W relationship from [10] with SFL from harvests, and estimated weights using SFL-A-W relationships (M3) from [11,12] with automatic system measurements (SFL and A).

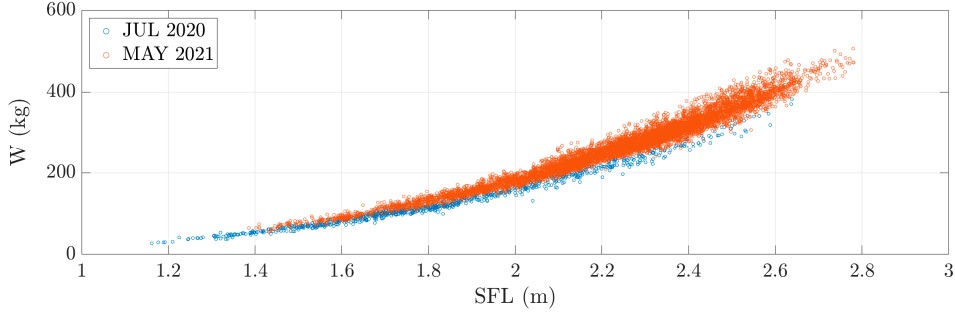

**Figure 11.** Estimated weight using SFL-A-W relationship (M3) from [11,12] with automatic system measurements in July 2020 and May 2021.

## 4. Conclusions and Further Work

The autonomous monitoring system operational from 28 July 2020 to 23 May 2021 in a fattening cage at Grup Balfegó (West Mediterranean) demonstrated its capacity to deliver thousands of accurate automatic measurements daily under ordinary circumstances using stereoscopic vision. This abundance of data enabled a detailed examination of tuna size evolution.

It was determined that analysing the evolution of the median and percentiles over time is inadequate for accurately estimating the growth of a fish population with varying ages and lengths. Instead, a modal analysis, employing Bhattacharya's method, was applied to identify different cohorts prior to analysing length and width evolution. The results indicated that from July 2020 to May 2021, the growth in length ranged approximately between 6 and 26 cm (2% to 21%), and the growth in width varied between 3.0 and 8.0 cm

(9% to 17%), depending on fish length. It was proven that, due to the special conditions in cages, both length and weight exhibited higher annual growth compared to the wild. Additionally, variability in fattening levels among fish of the same lengths was observed.

Moreover, utilising fish length and width enhanced the accuracy of weight estimation. Expressions derived from sacrificed fish in our earlier work [11,12] based on formulae relating weight and dimensions (length, width, and height) of BFT tuna fattened in captivity were validated. The results reinforced the notion that, for cage-farmed tuna, incorporating length and width dimensions refines weight estimation. While the automatic system was applied to size adult Atlantic bluefin tuna, the procedure could be adapted for other species by adjusting segmentation parameters, blob filtering criteria, and the geometric model.

As part of our future work, we intend to leverage deep learning techniques to expedite computing time, enabling the system to operate effectively in commercial operating environments. Additionally, new algorithms for fish tracking and counting will be developed. While this study provides initial positive results and conclusions regarding fish growth in cages, it underscores the need for a more complex model. This expanded model should delve into various aspects of growth, such as actual versus expected growth in the wild, growth linearity, the evolution of the condition factor, and the adaptation to changes in the imaging setup, water temperature, and turbidity. Such a comprehensive investigation requires a dedicated project with diverse setups, encompassing feeding regimes and fish ages, among other factors.

**Author Contributions:** Conceptualisation, G.A.-G., I.P.-A., V.E. and F.A.; methodology, P.M.-B., G.A.-G., I.P.-A. and V.E.; software, P.M.-B., J.M.-P., V.P.-P., A.M.-F. and V.A.-V.; validation, P.M.-B., V.P.-P., A.M.-F. and F.A.; formal analysis, P.M.-B., V.P.-P., A.M.-F. and F.A.; investigation, P.M.-B., J.M.-P., V.P.-P., A.M.-F. and P.O.-C.; resources, G.A.-G., P.O.-C., I.P.-A., V.E. and F.A.; data curation, P.M.-B., V.P.-P. and A.M.-F.; writing—original draft preparation, P.M.-B.; writing—review and editing, G.A.-G., V.E. and F.A.; visualisation, P.M.-B., V.P.-P., A.M.-F. and F.A.; supervision, G.A.-G., I.P.-A., V.E. and F.A.; project administration, G.A.-G. and V.E.; funding acquisition, G.A.-G., I.P.-A., V.E. and F.A. All authors have read and agreed to the published version of the manuscript.

**Funding:** This study forms part of the ThinkInAzul programme and was supported by the MCIN with funding from the European Union's NextGenerationEU (PRTR-C17.I1) and the Generalitat Valenciana (THINKINAZUL/2021/007, THINKINAZUL/2021/009, and AICO/2021/016). This work was carried out under the ICCAT Atlantic-Wide Research Programme for Bluefin Tuna (ICCAT GBYP 10/2020), which is funded by the European Union, several ICCAT CPCs, the ICCAT Secretariat, and other entities (see https://www.iccat.int/gbyp/en/overview.asp, accessed on 22 January 2024).

**Institutional Review Board Statement:** This study was conducted according to the guidelines of the Declaration of Helsinki. Ethical review and approval were waived for this study, because the use of underwater stereocameras to measure bluefin tuna dimensions is a non-invasive technique that does not cause them any suffering. The feeding activity is a normal activity in the commercial maintenance of bluefin tuna in cages.

**Informed Consent Statement:** Not applicable.

**Data Availability Statement:** The datasets presented in this article are not readily available because the data are part of an ongoing study. Requests to access the datasets should be directed to the corresponding author.

**Acknowledgments:** The content of this paper does not necessarily reflect ICCAT's point of view or that of any of the other sponsors, who carry no responsibility. In addition, it does not indicate the Commission's future policy in this area. This project has been possible thanks to the collaboration of Grup Balfegó. The autonomous monitoring system was designed and built in collaboration with Zunibal S.L.

**Conflicts of Interest:** The authors declare no conflicts of interest. The funders had no role in the design of this study; in the collection, analyses, or interpretation of data; in the writing of the manuscript; or in the decision to publish the results. Author Patricia Ordóñez-Cebrián is employed by Company Zunibal SL.

## Appendix A

**Table A1.** Identification of cohorts resulting from Bhattacharya's method for different periods from July 2020 to May 2021 with the automatic system from the ventral view. In grey and white, average SFL (cm); in blue, number of measurements (NM); in green, number of individuals (NI) according to the number of fish in the cage (724) for each cohort. * Cohorts with few specimens.

| Cohorts | | 1 * | 2 | 3 | 4 | 5 | 6 | 7 | 8 | 9 | 10 | 11 | Total Samples |
|---|---|---|---|---|---|---|---|---|---|---|---|---|---|
| 28–29 July | SFL | 122 | 136 | 156 | 171 | 183 | 200 | 214 | 226 | 237 | 249 | 266 | 925 |
| | NM | 13 | 54 | 166 | 181 | 141 | 156 | 61 | 82 | 43 | 24 | 4 | |
| | NI | 10 | 42 | 130 | 141 | 111 | 122 | 47 | 64 | 34 | 19 | 3 | |
| 8–11 August | SFL | 128 | 138 | 158 | 174 | 190 | 205 | 218 | 229 | 242 | 250 | 260 | 2011 |
| | NM | 37 | 152 | 371 | 458 | 298 | 229 | 271 | 102 | 70 | 13 | 10 | |
| | NI | 13 | 55 | 134 | 165 | 107 | 83 | 98 | 37 | 25 | 5 | 4 | |
| 23–26 September | SFL | | 149 | 165 | 181 | 195 | 213 | 227 | 238 | 249 | 262 | 272 | 7329 |
| | NM | | 23 | 46 | 179 | 206 | 341 | 547 | 383 | 301 | 185 | 30 | |
| | NI | | 7 | 15 | 58 | 67 | 110 | 177 | 124 | 97 | 60 | 10 | |
| 10–14 October | SFL | 139 | 150 | 167 | 183 | 196 | 212 | 226 | 239 | 251 | 262 | 272 | 14,538 |
| | NM | 32 | 263 | 422 | 1137 | 1032 | 2255 | 3267 | 2920 | 1867 | 1047 | 264 | |
| | NI | 2 | 13 | 21 | 57 | 51 | 113 | 163 | 146 | 93 | 52 | 13 | |
| 31 October–3 November | SFL | 150 | 154 | 175 | 187 | 199 | 213 | 224 | 236 | 248 | 259 | 268 | 5817 |
| | NM | 48 | 80 | 415 | 378 | 736 | 809 | 902 | 918 | 821 | 538 | 178 | |
| | NI | 6 | 10 | 52 | 47 | 91 | 101 | 112 | 114 | 102 | 67 | 22 | |
| 15–19 November | SFL | 141 | 157 | 174 | 187 | 200 | 213 | 225 | 236 | 249 | 258 | 268 | 10,468 |
| | NM | 19 | 159 | 369 | 739 | 837 | 1301 | 2066 | 1853 | 1754 | 932 | 402 | |
| | NI | 1 | 11 | 26 | 51 | 58 | 90 | 143 | 129 | 122 | 65 | 28 | |
| 6–8 December | SFL | 143 | 157 | 173 | 186 | 197 | 213 | 226 | 237 | 248 | 259 | 269 | 3851 |
| | NM | 7 | 86 | 96 | 266 | 255 | 662 | 574 | 704 | 645 | 367 | 152 | |
| | NI | 1 | 16 | 18 | 50 | 48 | 126 | 109 | 134 | 123 | 70 | 29 | |
| 15–18 December | SFL | 143 | 158 | 175 | 190 | 205 | 215 | 227 | 239 | 250 | 261 | 271 | 6305 |
| | NM | 7 | 91 | 178 | 315 | 422 | 468 | 826 | 587 | 671 | 355 | 93 | |
| | NI | 1 | 16 | 32 | 57 | 76 | 84 | 149 | 106 | 121 | 64 | 17 | |
| 21–22 January | SFL | | 152 | 170 | 182 | 197 | 209 | 221 | 235 | 246 | 256 | | 1664 |
| | NM | | 32 | 49 | 174 | 212 | 330 | 327 | 352 | 162 | 23 | | |
| | NI | | 14 | 21 | 76 | 92 | 144 | 142 | 153 | 70 | 10 | | |
| 15 February | SFL | 147 | 153 | 168 | 180 | 193 | 208 | 221 | 234 | 245 | 256 | | 1101 |
| | NM | 16 | 45 | 63 | 159 | 217 | 202 | 240 | 111 | 48 | 8 | | |
| | NI | 10 | 29 | 41 | 104 | 142 | 132 | 157 | 72 | 32 | 5 | | |
| 24–26 March | SFL | 143 | 158 | 176 | 189 | 203 | 216 | 227 | 239 | 250 | 261 | 270 | 20,942 |
| | NM | 83 | 431 | 1072 | 1692 | 2704 | 4004 | 4344 | 3095 | 2376 | 1031 | 100 | |
| | NI | 3 | 15 | 37 | 59 | 94 | 138 | 150 | 107 | 82 | 36 | 3 | |
| 5–7 May | SFL | 148 | 161 | 176 | 191 | 206 | 215 | 227 | 238 | 250 | 260 | 272 | 7518 |
| | NM | 58 | 163 | 318 | 607 | 1023 | 969 | 1773 | 1278 | 1043 | 244 | 35 | |
| | NI | 6 | 16 | 31 | 59 | 99 | 93 | 171 | 123 | 100 | 23 | 3 | |
| 21–23 May | SFL | 148 | 163 | 180 | 194 | 208 | 219 | 229 | 241 | 252 | 263 | | 4009 |
| | NM | 197 | 467 | 442 | 755 | 549 | 731 | 486 | 235 | 65 | 197 | | |
| | NI | 15 | 36 | 84 | 80 | 136 | 99 | 132 | 88 | 42 | 12 | | |
| Growth in length | | 26 | 27 | 24 | 23 | 25 | 19 | 15 | 15 | 15 | 14 | - | |

**Table A2.** Evolution of average SFL (cm) for each cohort identified with Bhattacharya's method. Cohorts 1 and 11 with few specimens have been omitted.

| | 28 July | 8 August | 23 September | 4 October | 31 October | 15 November | 6 December | 15 December | 21 January | 15 February | 24 March | 5 May | 21 May |
|---|---|---|---|---|---|---|---|---|---|---|---|---|---|
| Cohort #2 SFL (cm) | 136 | 138 | 149 | 150 | 154 | 157 | 157 | 158 | 152 | 153 | 158 | 161 | 161 |
| Cohort #3 SFL (cm) | 156 | 158 | 165 | 167 | 175 | 174 | 173 | 175 | 170 | 168 | 176 | 177 | 176 |
| Cohort #4 SFL (cm) | 171 | 174 | 181 | 183 | 187 | 187 | 186 | 190 | 182 | 180 | 189 | 190 | 191 |
| Cohort #5 SFL (cm) | 183 | 190 | 195 | 196 | 199 | 200 | 197 | 205 | 197 | 193 | 203 | 203 | 206 |
| Cohort #6 SFL (cm) | 200 | 205 | 213 | 212 | 213 | 213 | 213 | 215 | 209 | 208 | 216 | 212 | 215 |

**Table A2.** *Cont.*

| | 28 July | 8 August | 23 September | 4 October | 31 October | 15 November | 6 December | 15 December | 21 January | 15 February | 24 March | 5 May | 21 May |
|---|---|---|---|---|---|---|---|---|---|---|---|---|---|
| Cohort #7 SFL (cm) | 214 | 218 | 227 | 226 | 224 | 225 | 226 | 227 | 221 | 221 | 227 | 224 | 227 |
| Cohort #8 SFL (cm) | 226 | 229 | 238 | 239 | 236 | 236 | 237 | 239 | 235 | 234 | 239 | 236 | 238 |
| Cohort #9 SFL (cm) | 237 | 242 | 249 | 251 | 248 | 249 | 248 | 250 | 246 | 245 | 250 | 247 | 250 |
| Cohort #10 SFL (cm) | 249 | 250 | 262 | 262 | 259 | 258 | 259 | 261 | 256 | 256 | 261 | 259 | 260 |

**Table A3.** Evolution of average width ($\overline{A}$) from July 2020 to May 2021. In grey and white, average widths ($\overline{A}$) and their increase ($\overline{A}+$) in cm for fish grouped according to their straight fork length (SFL); in blue, number of samples of each cohort. * Cohorts with few samples.

| SFL Range | July 2020 | | August 2020 | | September 2020 | | December 2020 | | March 2021 | | May 2021 | | July 2020–May 2021 |
|---|---|---|---|---|---|---|---|---|---|---|---|---|---|
| | $\overline{A}$ | NM | $\overline{A}$ | NM | $\overline{A}$ | NM | $\overline{A}$ | NM | $\overline{A}$ | NM | $\overline{A}$ | NM | $\overline{A}$ |
| [140,150] | 28.2 | 46 | 29.0 | 89 | 29.3 | 45 | 30.2 | 11 | 31.0 | 84 | 32.1 | 36 | 3.9 (14%) |
| [150,160] | 29.5 | 75 | 31.1 | 171 | 31.8 | 46 | 32.7 | 59 | 33.7 | 254 | 33.5 | 69 | 4.0 (14%) |
| [160,170] | 32.3 | 113 | 33.0 | 252 | 33.3 | 136 | 35.0 | 38 | 35.7 | 264 | 35.3 | 119 | 3.0 (9%) |
| [170,180] | 33.6 | 128 | 34.2 | 253 | 36.2 | 223 | 36.4 | 89 | 38.2 | 656 | 38.2 | 232 | 4.6 (14%) |
| [180,190] | 34.6 | 115 | 36.1 | 215 | 37.4 | 421 | 38.8 | 179 | 40.6 | 1204 | 40.2 | 307 | 5.6 (16%) |
| [190,200] | 37.3 | 72 | 38.0 | 171 | 39.5 | 453 | 40.6 | 236 | 42.6 | 1486 | 42.5 | 472 | 5.2 (14%) |
| [200,210] | 38.8 | 73 | 39.9 | 187 | 41.4 | 450 | 42.1 | 255 | 44.9 | 1845 | 44.4 | 619 | 5.6 (14%) |
| [210,220] | 40.9 | 67 | 42.0 | 179 | 44.3 | 824 | 44.6 | 437 | 47.3 | 3134 | 47.0 | 1031 | 6.1 (15%) |
| [220,230] | 42.5 | 65 | 44.1 | 159 | 46.5 | 1203 | 47.2 | 536 | 49.4 | 3615 | 49.5 | 1316 | 7.0 (16%) |
| [230,240] | 44.0 | 46 | 46.0 | 67 | 48.7 | 1082 | 49.7 | 573 | 51.6 | 2986 | 51.1 | 1318 | 7.1 (16%) |
| [240,250] | 45.6 | 23 | 48.1 | 57 | 50.9 | 972 | 51.2 | 570 | 53.8 | 2574 | 53.3 | 938 | 7.7 (17%) |
| [250,260] | 47.2 | 13 | 49.4 | 17 | 53.5 | 715 | 53.9 | 454 | 55.8 | 1550 | 55.2 | 630 | 8.0 (17%) |
| [260,270] | 50.5 | 2 * | 52.1 | 4 * | 55.4 | 451 | 55.6 | 233 | 57.3 | 650 | 56.6 | 184 | 6.1 (12%) * |
| [270,280] | | | | | 56.7 | 110 | 58.4 | 64 | 58.5 | 93 | 57.9 | 33 | - |

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
