# Peer review of "Automated Monitoring of Bluefin Tuna Growth in Cages Using a Cohort-Based Approach"

_fishes, doi:10.3390/fishes9020046_

Round 1

Reviewer 1 Report

Comments and Suggestions for Authors

In this study, an automatic procedure is used to estimate the length and width of individuals in stereoscopic images of fish and its evolution over time is analyzed. And the following conclusions are drawn. Fish experience accelerated growth in cages compared to the wild, as demonstrated by applying a modal analysis (Bhattacharya’s method) to identify the cohorts of the population. Using a length-width-weight relationship to estimate fish weight gives more accurate results than traditional length-weight relationships. The study requires some clarification and improvements before being accepted for publication. Please find the comments below.

Suggestions and Questions:

1. In the introduction, the author should emphasize the significance of the study

2. Does the monitoring system have lighting equipment, and how to collect data when the light is dark?

3. The author wrote in section 2.2 that a computer vision algorithm is used to estimate the length and width of fish. I would like to know what the specific algorithm is used, can you give the details of the algorithm and how the length and width of fish are calculated?

4. How is the length of the fish calculated when its body is rotating?

5. What is the accuracy of calculating the weight of the fish using formula (1) and formula (2)?

6. Can the system be used to measure other types of fish?

Comments on the Quality of English Language

The Quality of English Language could be improved.

Reviewer 2 Report

Comments and Suggestions for Authors

Review of manuscript “Automated Bluefin Tuna growth monitoring in cages from a ventral perspective”

General comments:  This manuscript principally describes the application of a cohort identification approach (Bhattacharya’s method) to length estimates of free-swimming fish in an aquaculture setting.  It succeeds in demonstrating cohort-based intra-annual growth patterns that appear to exceed growth in the wild.  The addition of width to the LW relationship could be useful in improving accuracy for the aquaculture setting, but may not be as useful in monitoring and assessing wild populations.  There are some minor issues with English language use throughout.

The dataset is fairly large and adequate for yielding results on fish growth.  The biggest issue with the manuscript for me is I would like to see a more complex model to look at different aspects of growth in length and width thought the observation cycle, to be applied following the cohort identification process.  This modeling framework could also deal with the change in the imaging setup in the middle of the observation cycle, which is mentioned in the text but that should maybe be dealt with in a more quantitative manner.  In theory, stereo-calibration should create an equivalency between narrower and wider view image sets, but in reality, there is likely to be some effect which may include additional errors due to higher density, occlusions, higher distortion, etc.  A more robust way of dealing with this would be to actually fit a monthly (or daily) growth model to all the data, which can include a view angle effect factor as well as using the cohort as a random effect.  This would then replace the tendency lines in figure 5 (not sure how those are derived, linear regression?).  Additional covariates, including mean temperature, or visibility estimates (or Z perhaps), could be evaluated to explain some of the residual patterns.  The same model could be used then to quantify the amount of “excess growth” relative to expected grown in the wild, and could also be fit to fish width, or more appropriately width/length ratio. 

Several other questions could also be answered by a model, such as the linearity of growth, e.g. is there a strong start to growth in length in the beginning of the period, followed by a plateauing in the latter stages of the yearly cycle indicating that the excess growth rates hit some biological limits? This would require a fairly substantial re-write of the second component of the analysis, but in my opinion would yield a much richer and more robust set of results. 

I also think that expressing the width factor as a ratio of length (index of fatness?) would be a straightforward metric, especially given that the change in width appears to be proportionally similar across the length groupings.  This would remove the need for the length groups and allow a model to be fit to this that same as to the length data to possibly make better separation of the fattening vs growing aspects of the operation.   

Minor comments

Line 92 – the term “arduous” may be a bit strong for an automated procedure, complex perhaps?

Figure 1.  Please label the individual instrument components in the figure

Figure 4.  Why are not all length periods shown here (e.g. July and August 2020)?

Figure 5.  A table and graph are probably not both necessary.

Same applies to table 2 and figure 8

Consider a title change to include cohort, suggestion would be “Bluefin Tuna growth monitoring in cages using a cohort-based approach”

Comments on the Quality of English Language

There are some minor grammatical issues with English language use throughout the manuscript.

Round 2

Reviewer 1 Report

Comments and Suggestions for Authors

The question I raised has been revised

Comments on the Quality of English Language

The language of the paper is good, but some colloquial language needs improvement

Reviewer 2 Report

Comments and Suggestions for Authors

I thank the Authors for their quick work on revisions.  I look forward to their future study results.